# Uplift Modeling from Separate Labels

**Ikko Yamane**[1,2]    **Florian Yger**[3,2]    **Jamal Atif**[3]    **Masashi Sugiyama**[2,1]

[1] The University of Tokyo, CHIBA, JAPAN
[2] RIKEN Center for Advanced Intelligence Project (AIP), TOKYO, JAPAN
[3] LAMSADE, CNRS, Université Paris-Dauphine, Université PSL, PARIS, FRANCE
{yamane@ms., sugi@}k.u-tokyo.ac.jp, {florian.yger@, jamal.atif@}dauphine.fr

## Abstract

*Uplift modeling* is aimed at estimating the incremental impact of an action on an individual's behavior, which is useful in various application domains such as targeted marketing (advertisement campaigns) and personalized medicine (medical treatments). Conventional methods of uplift modeling require every instance to be *jointly* equipped with two types of labels: the taken action and its outcome. However, obtaining two labels for each instance at the same time is difficult or expensive in many real-world problems. In this paper, we propose a novel method of uplift modeling that is applicable to a more practical setting where only one type of labels is available for each instance. We show a mean squared error bound for the proposed estimator and demonstrate its effectiveness through experiments.

## 1    Introduction

In many real-world problems, a central objective is to optimally choose a right action to maximize the profit of interest. For example, in marketing, an advertising campaign is designed to promote people to purchase a product [29]. A marketer can choose whether to deliver an advertisement to each individual or not, and the outcome is the number of purchases of the product. Another example is personalized medicine, where a treatment is chosen depending on each patient to maximize the medical effect and minimize the risk of adverse events or harmful side effects [1, 13]. In this case, giving or not giving a medical treatment to each individual are the possible actions to choose, and the outcome is the rate of recovery or survival from the disease. Hereafter, we use the word *treatment* for taking an action, following the personalized medicine example.

*A/B testing* [14] is a standard method for such tasks, where two groups of people, A and B, are randomly chosen. The outcomes are measured separately from the two groups after treating all the members of Group A but none of Group B. By comparing the outcomes between the two groups by a statistical test, one can examine whether the treatment positively or negatively affected the outcome. However, A/B testing only compares the two extreme options: treating everyone or no one. These two options can be both far from optimal when the treatment has positive effect on some individuals but negative effect on others.

To overcome the drawback of A/B testing, *uplift modeling* has been investigated recently [11, 28, 32]. *Uplift modeling* is the problem of estimating the *individual uplift*, the incremental profit brought by the treatment conditioned on features of each individual. Uplift modeling enables us to design a refined decision rule for optimally determining whether to treat each individual or not, depending on his/her features. Such a treatment rule allows us to only target those who positively respond to the treatment and avoid treating negative responders.

In the standard uplift modeling setup, there are two types of labels [11, 28, 32]: One is whether the treatment has been given to the individual and the other is its outcome. Existing uplift modeling methods require each individual to be *jointly* given these two labels for analyzing the association

between outcomes and the treatment [11, 28, 32]. However, joint labels are expensive or hard (or even impossible) to obtain in many real-world problems. For example, when distributing an advertisement by email, we can easily record to whom the advertisement has been sent. However, for technical or privacy reasons, it is difficult to keep track of those people until we observe the outcomes on whether they buy the product or not. Alternatively, we can easily obtain information about purchasers of the product at the moment when the purchases are actually made. However, we cannot know whether those who are buying the product have been exposed to the advertisement or not. Thus, every individual always has one missing label. We term such samples *separately labeled samples*.

In this paper, we consider a more practical uplift modeling setup where no jointly labeled samples are available, but only separately labeled samples are given. Theoretically, we first show that the individual uplift is identifiable when we have two sets of separately labeled samples collected under *different* treatment policies. We then propose a novel method that directly estimates the individual uplift only from separately labeled samples. Finally, we demonstrate the effectiveness of the proposed method through experiments.

## 2 Problem Setting

This paper focuses on estimation of the *individual uplift* $u(\boldsymbol{x})$, often called *individual treatment effect (ITE)* in the causal inference literature [31], defined as $u(\boldsymbol{x}) := \mathbf{E}[Y_1 \mid \boldsymbol{x}] - \mathbf{E}[Y_{-1} \mid \boldsymbol{x}]$, where $\mathbf{E}[\,\cdot\mid\cdot\,]$ denotes the conditional expectation, and $\boldsymbol{x}$ is a $\mathcal{X}$-valued random variable ($\mathcal{X} \subseteq \mathbb{R}^d$) representing features of an individual, and $Y_1, Y_{-1}$ are $\mathcal{Y}$-valued *potential outcome variables* [31] ($\mathcal{Y} \subseteq \mathbb{R}$) representing outcomes that would be observed if the individual was treated and not treated, respectively. Note that only one of either $Y_1$ or $Y_{-1}$ can be observed for each individual. We denote the $\{1, -1\}$-valued random variable of the treatment assignment by $t$, where $t = 1$ means that the individual has been treated and $t = -1$ not treated. We refer to the population for which we want to evaluate $u(\boldsymbol{x})$ as the *test population*, and denote the density of the test population by $p(Y_1, Y_{-1}, \boldsymbol{x}, t)$.

We assume that $t$ is *unconfounded* with either of $Y_1$ and $Y_{-1}$ conditioned on $\boldsymbol{x}$, i.e. $p(Y_1 \mid \boldsymbol{x}, t) = p(Y_1 \mid \boldsymbol{x})$ and $p(Y_{-1} \mid \boldsymbol{x}, t) = p(Y_{-1} \mid \boldsymbol{x})$. Unconfoundedness is an assumption commonly made in observational studies [5, 33]. For notational convenience, we denote by $y := Y_t$ the outcome of the treatment assignment $t$. Furthermore, we refer to any conditional density of $t$ given $\boldsymbol{x}$ as a *treatment policy*.

In addition to the test population, we suppose that there are two *training populations* $k = 1, 2$, whose joint probability density $p_k(Y_1, Y_{-1}, \boldsymbol{x}, t)$ satisfy

$$p_k(Y_{t_0} = y_0 \mid \boldsymbol{x} = \boldsymbol{x}_0) = p(Y_{t_0} = y_0 \mid \boldsymbol{x} = \boldsymbol{x}_0) \quad \text{(for } k = 1, 2), \tag{1}$$

$$p_1(t = t_0 \mid \boldsymbol{x} = \boldsymbol{x}_0) \neq p_2(t = t_0 \mid \boldsymbol{x} = \boldsymbol{x}_0), \tag{2}$$

for all possible realizations $\boldsymbol{x}_0 \in \mathcal{X}$, $t_0 \in \{-1, 1\}$, and $y_0 \in \mathcal{Y}$. Intuitively, Eq. (1) means that potential outcomes depend on $\boldsymbol{x}$ in the same way as those in the test population, and Eq. (2) states that those two policies give a treatment with different probabilities for every $\boldsymbol{x} = \boldsymbol{x}_0$.

We suppose that the following four training data sets, which we call *separately labeled samples*, are given:

$$\{(\boldsymbol{x}_i^{(k)}, y_i^{(k)})\}_{i=1}^{n_k} \overset{\text{i.i.d.}}{\sim} p_k(\boldsymbol{x}, y), \quad \{(\widetilde{\boldsymbol{x}}_i^{(k)}, t_i^{(k)})\}_{i=1}^{\widetilde{n}_k} \overset{\text{i.i.d.}}{\sim} p_k(\boldsymbol{x}, t) \quad \text{(for } k = 1, 2),$$

where $n_k$ and $\widetilde{n}_k$, $k = 1, 2$, are positive integers. Under Assumptions (1), (2), and the uncon-foundedness, we have $p_k(Y_t \mid \boldsymbol{x}, t = t_0) = p(Y_{t_0} \mid \boldsymbol{x}, t = t_0) = p(Y_{t_0} \mid \boldsymbol{x})$ for $t_0 \in \{-1, 1\}$ and $k \in \{1, 2\}$. Note that we can safely denote $p(y \mid \boldsymbol{x}, t) := p_k(y \mid \boldsymbol{x}, t)$. Moreover, we have $\mathbf{E}[Y_{t_0} \mid \boldsymbol{x}] = \mathbf{E}[y \mid \boldsymbol{x}, t = t_0]$ for $t_0 = 1, -1$, and thus our goal boils down to the estimation of

$$u(\boldsymbol{x}) = \mathbf{E}[y \mid \boldsymbol{x}, t = 1] - \mathbf{E}[y \mid \boldsymbol{x}, t = -1] \tag{3}$$

from the separately labeled samples, where the conditional expectation is taken over $p(y \mid \boldsymbol{x}, t)$.

Estimation of the individual uplift is important for the following reasons.

**It enables the estimation of the average uplift.** The *average uplift* $U(\pi)$ of the treatment policy $\pi(t \mid \boldsymbol{x})$ is the average outcome of $\pi$, subtracted by that of the policy $\pi_-$, which constantly assigns

the treatment as $t = -1$, i.e., $\pi_-(t = \tau \mid \boldsymbol{x}) := 1[\tau = -1]$, where $1[\cdot]$ denotes the indicator function:

$$U(\pi) := \iint \sum_{t=-1,1} yp(y \mid \boldsymbol{x}, t)\pi(t \mid \boldsymbol{x})p(\boldsymbol{x})\mathrm{d}y\mathrm{d}\boldsymbol{x} - \iint \sum_{t=-1,1} yp(y \mid \boldsymbol{x}, t)\pi_-(t \mid \boldsymbol{x})p(\boldsymbol{x})\mathrm{d}y\mathrm{d}\boldsymbol{x}$$

$$= \int u(\boldsymbol{x})\pi(t = 1 \mid \boldsymbol{x})p(\boldsymbol{x})\mathrm{d}\boldsymbol{x}. \tag{4}$$

This quantity can be estimated from samples of $\boldsymbol{x}$ once we obtain an estimate of $u(\boldsymbol{x})$.

**It provides the optimal treatment policy.** The treatment policy given by $\pi(t = 1 \mid \boldsymbol{x}) = 1[0 \leq u(\boldsymbol{x})]$ is the optimal treatment that maximizes the average uplift $U(\pi)$ and equivalently the average outcome $\iint \sum_{t=-1,1} yp(y \mid \boldsymbol{x}, t)\pi(t \mid \boldsymbol{x})p(\boldsymbol{x})\mathrm{d}y\mathrm{d}\boldsymbol{x}$ (see Eq. (4)) [32].

**It is the optimal ranking scoring function.** From a practical viewpoint, it may be useful to prioritize individuals to be treated according to some ranking scores especially when the treatment is costly and only a limited number of individuals can be treated due to some budget constraint. In fact, $u(\boldsymbol{x})$ serves as the optimal ranking scores for this purpose [36]. More specifically, we define a family of treatment policies $\{\pi_{f,\alpha}\}_{\alpha \in \mathbb{R}}$ *associated with scoring function* $f$ by $\pi_{f,\alpha}(t = 1 \mid \boldsymbol{x}) = 1[\alpha \leq f(\boldsymbol{x})]$. Then, under some technical condition, $f = u$ maximizes the *area under the uplift curve (AUUC)* defined as

$$\mathrm{AUUC}(f) := \int_0^1 U(\pi_{f,\alpha})\mathrm{d}C_\alpha$$

$$= \int_0^1 \int u(\boldsymbol{x})1[\alpha \leq f(\boldsymbol{x})]p(\boldsymbol{x})\mathrm{d}\boldsymbol{x}\mathrm{d}C_\alpha$$

$$= \mathbf{E}[1[f(\boldsymbol{x}) \leq f(\boldsymbol{x}')]u(\boldsymbol{x}')],$$

where $C_\alpha := \Pr[f(\boldsymbol{x}) < \alpha]$, $\boldsymbol{x}, \boldsymbol{x}' \overset{\text{i.i.d.}}{\sim} p(\boldsymbol{x})$, and $\mathbf{E}$ denotes the expectation with respect to these variables. AUUC is a standard performance measure for uplift modeling methods [11, 25, 28, 32]. For more details, see Appendix B in the supplementary material.

**Remark on the problem setting:** Uplift modeling is often referred to as individual treatment effect estimation or heterogeneous treatment effect estimation and has been extensively studied especially in the causal inference literature [5, 7, 9, 12, 16, 24, 31, 37]. In particular, recent research has investigated the problem under the setting of *observational studies*, inference using data obtained from *uncontrolled experiments* because of its practical importance [33]. Here, experiments are said to be uncontrolled when some of treatment variables are not controlled to have designed values.

Given that treatment policies are unknown, our problem setting is also of observational studies but poses an additional challenge that stems from missing labels. What makes our problem feasible is that we have two kinds of data sets following different treatment policies.

It is also important to note that our setting generalizes the standard setting for observational studies since the former is reduced to the latter when one of the treatment policies always assigns individuals to the treatment group, and the other to the control group.

Our problem is also closely related to individual treatment effect estimation via instrumental variables [2, 6, 10, 19].[1]

## 3   Naive Estimators

A naive approach is first estimating the conditional density $p_k(y \mid \boldsymbol{x})$ and $p_k(t \mid \boldsymbol{x})$ from training samples by some conditional density estimator [4, 34], and then solving the following linear system for $p(y \mid \boldsymbol{x}, t = 1)$ and $p(y \mid \boldsymbol{x}, t = -1)$:

$$\underbrace{p_k(y \mid \boldsymbol{x})}_{\text{Estimated from } \{(\boldsymbol{x}_i^{(k)}, y_i^{(k)})\}_{i=1}^n} = \sum_{t=-1,1} p(y \mid \boldsymbol{x}, t) \underbrace{p_k(t \mid \boldsymbol{x})}_{\text{Estimated from } \{(\widetilde{\boldsymbol{x}}_i^{(k)}, t_i^{(k)})\}_{i=1}^{\widetilde{n}}} \quad \text{(for } k = 1, 2). \tag{5}$$

After that, the conditional expectations of $y$ over $p(y \mid \boldsymbol{x}, t = 1)$ and $p(y \mid \boldsymbol{x}, t = -1)$ are calculated by numerical integration, and finally their difference is calculated to obtain another estimate of $u(\boldsymbol{x})$.

However, this may not yield a good estimate due to the difficulty of conditional density estimation and the instability of numerical integration. This issue may be alleviated by working on the following linear system implied by Eq. (5) instead: $\mathbf{E}_k[y \mid \boldsymbol{x}] = \sum_{t=-1,1} \mathbf{E}[y \mid \boldsymbol{x}, t] p_k(t \mid \boldsymbol{x})$, $k = 1, 2$, where $\mathbf{E}_k[y \mid \boldsymbol{x}]$ and $p_k(t \mid \boldsymbol{x})$ can be estimated from our samples. Solving this new system for $\mathbf{E}[y \mid \boldsymbol{x}, t = 1]$ and $\mathbf{E}[y \mid \boldsymbol{x}, t = -1]$ and taking their difference gives an estimate of $u(\boldsymbol{x})$. A method called *two-stage least-squares* for instrumental variable regression takes such an approach [10].

The second approach of estimation $E_k[y|x]$ and $p_k(t|x)$ avoids both conditional density estimation and numerical integration, but it still involves post processing of solving the linear system and subtraction, being a potential cause of performance deterioration.

# 4    Proposed Method

In this section, we develop a method that can overcome the aforementioned problems by directly estimating the individual uplift.

## 4.1    Direct Least-Square Estimation of the Individual Uplift

First, we will show an important lemma that directly relates the marginal distributions of separately labeled samples to the individual uplift $u(\boldsymbol{x})$.

**Lemma 1.** *For every $\boldsymbol{x}$ such that $p_1(\boldsymbol{x}) \neq p_2(\boldsymbol{x})$, $u(\boldsymbol{x})$ can be expressed as*

$$u(\boldsymbol{x}) = 2 \times \frac{\mathbf{E}_{y \sim p_1(y|\boldsymbol{x})}[y] - \mathbf{E}_{y \sim p_2(y|\boldsymbol{x})}[y]}{\mathbf{E}_{t \sim p_1(t|\boldsymbol{x})}[t] - \mathbf{E}_{t \sim p_2(t|\boldsymbol{x})}[t]}. \tag{6}$$

For a proof, refer to Appendix C in the supplementary material.

Using Eq. (6), we can re-interpret the naive methods described in Section 3 as estimating the conditional expectations on the right-hand side by separately performing regression on $\{(\boldsymbol{x}_i^{(1)}, y_i^{(1)})\}_{i=1}^{n_1}$, $\{(\boldsymbol{x}_i^{(2)}, y_i^{(2)})\}_{i=1}^{n_2}$, $\{(\widetilde{\boldsymbol{x}}_i^{(1)}, t_i^{(1)})\}_{i=1}^{\widetilde{n}_1}$, and $\{(\widetilde{\boldsymbol{x}}_i^{(2)}, t_i^{(2)})\}_{i=1}^{\widetilde{n}_2}$. This approach may result in unreliable performance when the denominator is close to zero, i.e., $p_1(t \mid \boldsymbol{x}) \simeq p_2(t \mid \boldsymbol{x})$.

Lemma 1 can be simplified by introducing auxiliary variables $z$ and $w$, which are $\mathcal{Z}$-valued and $\{-1, 1\}$-valued random variables whose conditional probability density and mass are defined by

$$p(z = z_0 \mid \boldsymbol{x}) = \tfrac{1}{2} p_1(y = z_0 \mid \boldsymbol{x}) + \tfrac{1}{2} p_2(y = -z_0 \mid \boldsymbol{x}),$$
$$p(w = w_0 \mid \boldsymbol{x}) = \tfrac{1}{2} p_1(t = w_0 \mid \boldsymbol{x}) + \tfrac{1}{2} p_2(t = -w_0 \mid \boldsymbol{x}),$$

for any $z_0 \in \mathcal{Z}$ and any $w_0 \in \{-1, 1\}$, where $\mathcal{Z} := \{s_0 y_0 \mid y_0 \in \mathcal{Y}, s_0 \in \{1, -1\}\}$.

**Lemma 2.** *For every $\boldsymbol{x}$ such that $p_1(\boldsymbol{x}) \neq p_2(\boldsymbol{x})$, $u(\boldsymbol{x})$ can be expressed as*

$$u(\boldsymbol{x}) = 2 \times \frac{\mathbf{E}[z \mid \boldsymbol{x}]}{\mathbf{E}[w \mid \boldsymbol{x}]},$$

*where $\mathbf{E}[z \mid \boldsymbol{x}]$ and $\mathbf{E}[w \mid \boldsymbol{x}]$ are the conditional expectations of $z$ given $\boldsymbol{x}$ over $p(z \mid \boldsymbol{x})$ and $w$ given $\boldsymbol{x}$ over $p(w \mid \boldsymbol{x})$, respectively.*

A proof can be found in Appendix D in the supplementary material.

Let $w_i^{(k)} := (-1)^{k-1} t_i^{(k)}$ and $z_i^{(k)} := (-1)^{k-1} y_i^{(k)}$. Assuming that $p_1(\boldsymbol{x}) = p_2(\boldsymbol{x}) =: p(\boldsymbol{x})$, $n_1 = n_2$, and $\widetilde{n}_1 = \widetilde{n}_2$ for simplicity, $\{(\widetilde{\boldsymbol{x}}_i, w_i)\}_{i=1}^n := \{(\widetilde{\boldsymbol{x}}_i^{(k)}, w_i^{(k)})\}_{k=1,2; \, i=1,\dots,\widetilde{n}_k}$ and $\{(\boldsymbol{x}_i, z_i)\}_{i=1}^n := \{(\boldsymbol{x}_i^{(k)}, z_i^{(k)})\}_{k=1,2; \, i=1,\dots,n_k}$ can be seen as samples drawn from $p(\boldsymbol{x}, z) := p(z \mid \boldsymbol{x}) p(\boldsymbol{x})$ and $p(\boldsymbol{x}, w) := p(w \mid \boldsymbol{x}) p(\boldsymbol{x})$, respectively, where $n = n_1 + n_2$ and $\widetilde{n} = \widetilde{n}_1 + \widetilde{n}_2$. The more general cases where $p_1(\boldsymbol{x}) \neq p_2(\boldsymbol{x})$, $n_1 \neq n_2$, or $\widetilde{n}_1 \neq \widetilde{n}_2$ are discussed in Appendix I in the supplementary material.

**Theorem 1.** *Assume that $\mu_w, \mu_z \in L^2(p)$ and $\mu_w(\boldsymbol{x}) \neq 0$ for every $\boldsymbol{x}$ such that $p(\boldsymbol{x}) > 0$, where $L^2(p) := \{f : \mathcal{X} \to \mathbb{R} \mid \mathbf{E}_{\boldsymbol{x} \sim p(\boldsymbol{x})}[f(\boldsymbol{x})^2] < \infty\}$. The individual uplift $u(\boldsymbol{x})$ equals the solution to the following least-squares problem:*

$$u(\boldsymbol{x}) = \operatorname*{argmin}_{f \in L^2(p)} \mathbf{E}[(\mu_w(\boldsymbol{x})f(\boldsymbol{x}) - 2\mu_z(\boldsymbol{x}))^2], \tag{7}$$

*where $\mathbf{E}$ denotes the expectation over $p(\boldsymbol{x})$, $\mu_w(\boldsymbol{x}) := \mathbf{E}[w \mid \boldsymbol{x}]$, and $\mu_z(\boldsymbol{x}) := \mathbf{E}[z \mid \boldsymbol{x}]$.*

Theorem 1 follows from Lemma 2. Note that $p_1(\boldsymbol{x}) \neq p_2(\boldsymbol{x})$ in Eq. (2) implies $\mu_w(\boldsymbol{x}) \neq 0$.

In what follows, we develop a method that directly estimates $u(\boldsymbol{x})$ by solving Eq. (7). A challenge here is that it is not straightforward to evaluate the objective functional since it involves unknown functions, $\mu_w$ and $\mu_z$.

## 4.2 Disentanglement of $z$ and $w$

Our idea is to transform the objective functional in Eq. (7) into another form in which $\mu_w(\boldsymbol{x})$ and $\mu_z(\boldsymbol{x})$ appear separately and linearly inside the expectation operator so that we can approximate them using our separately labeled samples.

For any function $g \in L^2(p)$ and any $\boldsymbol{x} \in \mathcal{X}$, expanding the left-hand side of the inequality $\mathbf{E}[(\mu_w(\boldsymbol{x})f(\boldsymbol{x}) - 2\mu_z(\boldsymbol{x}) - g(\boldsymbol{x}))^2] \geq 0$, we have

$$\mathbf{E}[(\mu_w(\boldsymbol{x})f(\boldsymbol{x}) - 2\mu_z(\boldsymbol{x}))^2] \geq 2\mathbf{E}[(\mu_w(\boldsymbol{x})f(\boldsymbol{x}) - 2\mu_z(\boldsymbol{x}))g(\boldsymbol{x})] - \mathbf{E}[g(\boldsymbol{x})^2] =: J(f,g). \tag{8}$$

The equality is attained when $g(\boldsymbol{x}) = \mu_w(\boldsymbol{x})f(\boldsymbol{x}) - \mu_z(\boldsymbol{x})$ for any fixed $f$. This means that the objective functional of Eq. (7) can be calculated by maximizing $J(f,g)$ with respect to $g$. Hence,

$$u(\boldsymbol{x}) = \operatorname*{argmin}_{f \in L^2(p)} \max_{g \in L^2(p)} J(f,g). \tag{9}$$

Furthermore, $\mu_w$ and $\mu_z$ are separately and linearly included in $J(f,g)$, which makes it possible to write it in terms of $z$ and $w$ as

$$J(f,g) = 2\mathbf{E}[wf(\boldsymbol{x})g(\boldsymbol{x})] - 4\mathbf{E}[zg(\boldsymbol{x})] - \mathbf{E}[g(\boldsymbol{x})^2]. \tag{10}$$

Unlike the original objective functional in Eq. (7), $J(f,g)$ can be easily estimated using sample averages by

$$\widehat{J}(f,g) = \frac{2}{\widetilde{n}} \sum_{i=1}^{\widetilde{n}} w_i f(\widetilde{\boldsymbol{x}}_i) g(\widetilde{\boldsymbol{x}}_i) - \frac{4}{n} \sum_{i=1}^{n} z_i g(\boldsymbol{x}_i) - \frac{1}{2n} \sum_{i=1}^{n} g(\boldsymbol{x}_i)^2 - \frac{1}{2\widetilde{n}} \sum_{i=1}^{\widetilde{n}} g(\widetilde{\boldsymbol{x}}_i)^2. \tag{11}$$

In practice, we solve the following regularized empirical optimization problem:

$$\min_{f \in F} \max_{g \in G} \widehat{J}(f,g) + \Omega(f,g), \tag{12}$$

where $F$, $G$ are models for $f$, $g$ respectively, and $\Omega(f,g)$ is some regularizer.

An advantage of the proposed framework is that it is model-independent, and any models can be trained by optimizing the above objective.

The function $g$ can be interpreted as a *critic* of $f$ as follows. Minimizing Eq. (10) with respect to $f$ is equivalent to minimizing $J(f,g) = \mathbf{E}[g(\boldsymbol{x})\{\mu_w(\boldsymbol{x})f(\boldsymbol{x}) - 2\mu_z(\boldsymbol{x})\}]$. $g(\boldsymbol{x})$ serves as a good critic of $f(\boldsymbol{x})$ when it makes the cost $g(\boldsymbol{x})\{\mu_w(\boldsymbol{x})f(\boldsymbol{x}) - 2\mu_z(\boldsymbol{x})\}$ larger for $\boldsymbol{x}$ at which $f$ makes a larger error $|\mu_w(\boldsymbol{x})f(\boldsymbol{x}) - 2\mu_z(\boldsymbol{x})|$. In particular, $g$ maximizes the objective above when $g(\boldsymbol{x}) = \mu_w(\boldsymbol{x})f(\boldsymbol{x}) - 2\mu_z(\boldsymbol{x})$ for any $f$, and the maximum coincides with the least-squares objective in Eq. (7).

Suppose that $F$ and $G$ are linear-in-parameter models: $F = \{f_{\boldsymbol{\alpha}} : \boldsymbol{x} \mapsto \boldsymbol{\alpha}^\top \boldsymbol{\phi}(\boldsymbol{x}) \mid \boldsymbol{\alpha} \in \mathbb{R}^{b_f}\}$ and $G = \{g_{\boldsymbol{\beta}} : \boldsymbol{x} \mapsto \boldsymbol{\beta}^\top \boldsymbol{\psi}(\boldsymbol{x}) \mid \boldsymbol{\beta} \in \mathbb{R}^{b_g}\}$, where $\boldsymbol{\phi}$ and $\boldsymbol{\psi}$ are $b_f$-dimensional and $b_g$-dimensional vectors of basis functions in $L^2(p)$. Then, $\widehat{J}(f_{\boldsymbol{\alpha}}, g_{\boldsymbol{\beta}}) = 2\boldsymbol{\alpha}^\top \boldsymbol{A}\boldsymbol{\beta} - 4\boldsymbol{b}^\top \boldsymbol{\beta} - \boldsymbol{\beta}^\top \boldsymbol{C}\boldsymbol{\beta}$, where

$$\boldsymbol{A} := \frac{1}{\widetilde{n}} \sum_{i=1}^{\widetilde{n}} w_i \boldsymbol{\phi}(\widetilde{\boldsymbol{x}}_i)\boldsymbol{\psi}(\widetilde{\boldsymbol{x}}_i)^\top, \quad \boldsymbol{b} := \frac{1}{n} \sum_{i=1}^{n} z_i \boldsymbol{\psi}(\boldsymbol{x}_i),$$

$$\boldsymbol{C} := \frac{1}{2n} \sum_{i=1}^{n} \boldsymbol{\psi}(\boldsymbol{x}_i)\boldsymbol{\psi}(\boldsymbol{x}_i)^\top + \frac{1}{2\widetilde{n}} \sum_{i=1}^{\widetilde{n}} \boldsymbol{\psi}(\widetilde{\boldsymbol{x}}_i)\boldsymbol{\psi}(\widetilde{\boldsymbol{x}}_i)^\top.$$

Using $\ell_2$-regularizers, $\Omega(f, g) = \lambda_{\mathrm{f}} \boldsymbol{\alpha}^\top \boldsymbol{\alpha} - \lambda_{\mathrm{g}} \boldsymbol{\beta}^\top \boldsymbol{\beta}$ with some positive constants $\lambda_{\mathrm{f}}$ and $\lambda_{\mathrm{g}}$, the solution to the inner maximization problem can be obtained in the following analytical form:

$$\widehat{\boldsymbol{\beta}}_{\boldsymbol{\alpha}} := \operatorname*{argmax}_{\boldsymbol{\beta}} \widehat{J}(f_{\boldsymbol{\alpha}}, g_{\boldsymbol{\beta}}) = \widetilde{\boldsymbol{C}}^{-1}(\boldsymbol{A}^\top \boldsymbol{\alpha} - 2\boldsymbol{b}),$$

where $\widetilde{\boldsymbol{C}} = \boldsymbol{C} + \lambda_{\mathrm{g}} \boldsymbol{I}_{b_{\mathrm{g}}}$ and $\boldsymbol{I}_{b_{\mathrm{g}}}$ is the $b_{\mathrm{g}}$-by-$b_{\mathrm{g}}$ identity matrix. Then, we can obtain the solution to Eq. (12) analytically as

$$\widehat{\boldsymbol{\alpha}} := \operatorname*{argmin}_{\boldsymbol{\alpha}} \widehat{J}(f_{\boldsymbol{\alpha}}, g_{\widehat{\boldsymbol{\beta}}_{\boldsymbol{\alpha}}}) = 2(\boldsymbol{A}\widetilde{\boldsymbol{C}}^{-1}\boldsymbol{A}^\top + \lambda_{\mathrm{f}} \boldsymbol{I}_{b_{\mathrm{g}}})^{-1}\boldsymbol{A}\widetilde{\boldsymbol{C}}^{-1}\boldsymbol{b}.$$

Finally, from Eq. (7), our estimate of $u(\boldsymbol{x})$ is given as $\widehat{\boldsymbol{\alpha}}^\top \boldsymbol{\phi}(\boldsymbol{x})$.

**Remark on model selection:** Model selection for $F$ and $G$ is not straightforward since the test performance measure cannot be directly evaluated with (held out) training data of our problem. Instead, we may evaluate the value of $J(\widehat{f}, \widehat{g})$, where $(\widehat{f}, \widehat{g}) \in F \times G$ is the optimal solution pair to $\min_{f \in F} \max_{g \in G} \widehat{J}(f, g)$. However, it is still nontrivial to tell if the objective value is small because the solution is good in terms of the outer minimization, or because it is poor in terms of the inner maximization. We leave this issue for future work.

## 5 Theoretical Analysis

A theoretically appealing property of the proposed method is that its objective consists of simple sample averages. This enables us to establish a generalization error bound in terms of the Rademacher complexity [15, 22].

Denote $\varepsilon_G(f) := \sup_{g \in L^2(p)} J(f, g) - \sup_{g \in G} J(f, g)$. Also, let $\mathfrak{R}_q^N(H)$ denote the *Rademacher complexity* of a set of functions $H$ over $N$ random variables following probability density $q$ (refer to Appendix E for the definition). Proofs of the following theorems and corollary can be found in Appendix E, Appendix F, and Appendix G in the supplementary material.

**Theorem 2.** *Assume that $n_1 = n_2$, $\widetilde{n}_1 = \widetilde{n}_2$, $p_1(\boldsymbol{x}) = p_2(\boldsymbol{x})$, $W := \inf_{x \in \mathcal{X}} |\mu_w(\boldsymbol{x})| > 0$, $M_{\mathcal{Z}} := \sup_{z \in \mathcal{Z}} |z| < \infty$, $M_F := \sup_{f \in F, x \in \mathcal{X}} |f(x)| < \infty$, and $M_G := \sup_{g \in G, x \in \mathcal{X}} |g(x)| < \infty$. Then, the following holds with probability at least $1 - \delta$ for every $f \in F$:*

$$\mathbf{E}_{\boldsymbol{x} \sim p(\boldsymbol{x})}[(f(\boldsymbol{x}) - u(\boldsymbol{x}))^2] \leq \frac{1}{W^2}\left[\sup_{g \in G} \widehat{J}(f, g) + \mathcal{R}_{F,G}^{n,\widetilde{n}} + \left(\frac{M_z}{\sqrt{2n}} + \frac{M_w}{\sqrt{2\widetilde{n}}}\right)\sqrt{\log \frac{2}{\delta}} + \varepsilon_G(f)\right],$$

*where* $M_z := 4M_{\mathcal{Y}}M_{\mathcal{G}} + M_{\mathcal{G}}^2/2$, $M_w = 2M_{\mathcal{F}}M_{\mathcal{G}} + M_{\mathcal{G}}^2/2$, *and* $\mathcal{R}_{F,G}^{n,\widetilde{n}} := 2(M_F + 4M_{\mathcal{Z}})\mathfrak{R}_{p(\boldsymbol{x},z)}^n(G) + 2(2M_F + M_G)\mathfrak{R}_{p(\boldsymbol{x},w)}^{\widetilde{n}}(F) + 2(M_F + M_G)\mathfrak{R}_{p(\boldsymbol{x},w)}^{\widetilde{n}}(G)$.

In particular, the following bound holds for the linear-in-parameter models.

**Corollary 1.** *Let $F = \{x \mapsto \boldsymbol{\alpha}^\top \boldsymbol{\phi}(\boldsymbol{x}) \mid \|\boldsymbol{\alpha}\|_2 \leq \Lambda_F\}$, $G = \{x \mapsto \boldsymbol{\beta}^\top \boldsymbol{\psi}(\boldsymbol{x}) \mid \|\boldsymbol{\beta}\|_2 \leq \Lambda_G\}$. Assume that $r_F := \sup_{\boldsymbol{x} \in \mathcal{X}} \|\boldsymbol{\phi}(\boldsymbol{x})\| < \infty$ and $r_G := \sup_{\boldsymbol{x} \in \mathcal{X}} \|\boldsymbol{\psi}(\boldsymbol{x})\| < \infty$, where $\|\cdot\|_2$ is the $L_2$-norm. Under the assumptions of Theorem 2, it holds with probability at least $1 - \delta$ that for every $f \in F$,*

$$\mathbf{E}_{\boldsymbol{x} \sim p(\boldsymbol{x})}[(f(\boldsymbol{x}) - u(\boldsymbol{x}))^2] \leq \frac{1}{W^2}\left[\sup_{g \in G} \widehat{J}(f, g) + \frac{C_z\sqrt{\log \frac{2}{\delta}} + D_z}{\sqrt{2n}} + \frac{C_w\sqrt{\log \frac{2}{\delta}} + D_w}{\sqrt{2\widetilde{n}}} + \varepsilon_G(f)\right],$$

*where* $C_z := r_G^2 \Lambda_G^2 + 4r_G \Lambda_G M_{\mathcal{Y}}$, $C_w := 2r_F^2 \Lambda_F^2 + 2r_F r_G \Lambda_F \Lambda_G + r_G^2 \Lambda_G^2$, $D_z := r_G^2 \Lambda_G^2/2 + 4r_G \Lambda_G M_{\mathcal{Y}}$, *and* $D_w := r_G^2 \Lambda_G^2/2 + 4r_F r_G \Lambda_F \Lambda_G$.

Theorem 2 and Corollary 1 imply that minimizing $\sup_{g \in G} \widehat{J}(f, g)$, as the proposed method does, amounts to minimizing an upper bound of the mean squared error. In fact, for the linear-in-parameter models, it can be shown that the mean squared error of the proposed estimator is upper bounded by $O(1/\sqrt{n} + 1/\sqrt{\widetilde{n}})$ plus some model mis-specification error with high probability as follows.

**Theorem 3** (Informal). *Let $\widehat{f} \in F$ be any approximate solution to $\inf_{f \in F} \sup_{g \in G} \widehat{J}(f, g)$ with sufficient precision. Under the assumptions of Corollary 1, it holds with probability at least $1 - \delta$ that*

$$\mathbf{E}_{\boldsymbol{x} \sim p(\boldsymbol{x})}[(\widehat{f}(\boldsymbol{x}) - u(\boldsymbol{x}))^2] \leq O\left(\left(\frac{1}{\sqrt{n}} + \frac{1}{\sqrt{\widetilde{n}}}\right) \log \frac{1}{\delta}\right) + \frac{2\varepsilon_G^F + \varepsilon_F}{W^2}, \tag{13}$$

*where $\varepsilon_G^F := \sup_{f \in F} \varepsilon_G(f)$ and $\varepsilon_F := \inf_{f \in F} J(f)$.*

A more formal version of Theorem 3 can be found in Appendix G.

## 6 More General Loss Functions

Our framework can be extended to more general loss functions:

$$\inf_{f \in L^2(p)} \mathbf{E}[\ell(\mu_w(\boldsymbol{x})f(\boldsymbol{x}), 2\mu_z(\boldsymbol{x}))], \tag{14}$$

where $\ell : \mathbb{R} \times \mathbb{R} \to \mathbb{R}$ is a loss function that is lower semi-continuous and convex with respect to both the first and the second arguments, where a function $\varphi : \mathbb{R} \to \mathbb{R}$ is *lower semi-continuous* if $\liminf_{y \to y_0} \varphi(y) = \varphi(y_0)$ for every $y_0 \in \mathbb{R}$ [30].[2] As with the squared loss, a major difficulty in solving this optimization problem is that the operand of the expectation has nonlinear dependency on both $\mu_w(\boldsymbol{x})$ and $\mu_z(\boldsymbol{x})$ at the same time. Below, we will show a way to transform the objective functional into a form that can be easily approximated using separately labeled samples.

From the assumptions on $\ell$, we have $\ell(y, y') = \sup_{z \in \mathbb{R}} yz - \ell^*(z, y')$, where $\ell^*(\cdot, y')$ is the convex conjugate of the function $y \mapsto \ell(y, y')$ defined for any $y' \in \mathbb{R}$ as $z \mapsto \ell^*(z, y') = \sup_{y \in \mathbb{R}}[yz - \ell(y, y')]$ (see Rockafellar [30]). Hence,

$$\mathbf{E}[\ell(\mu_w(\boldsymbol{x})f(\boldsymbol{x}), 2\mu_z(\boldsymbol{x}))] = \sup_{g \in L^2(p)} \mathbf{E}[\mu_w(\boldsymbol{x})f(\boldsymbol{x})g(\boldsymbol{x}) - \ell^*(g(\boldsymbol{x}), 2\mu_z(\boldsymbol{x}))].$$

Similarly, we obtain $\mathbf{E}[\ell^*(g(\boldsymbol{x}), 2\mu_z(\boldsymbol{x}))] = \sup_{h \in L^2(p)} 2\mathbf{E}[\mu_z(\boldsymbol{x})h(\boldsymbol{x})] - \mathbf{E}[\ell_*^*(g(\boldsymbol{x}), h(\boldsymbol{x}))]$, where $\ell_*^*(y, \cdot)$ is the convex conjugate of the function $y' \mapsto \ell^*(y, y')$ defined for any $y, z' \in \mathbb{R}$ by $\ell_*^*(y, z') := \sup_{y' \in \mathbb{R}}[y'z - \ell^*(y, y')]$. Thus, Eq. (14) can be rewritten as

$$\inf_{f \in L^2(p)} \sup_{g \in L^2(p)} \inf_{h \in L^2(p)} K(f, g, h),$$

where $K(f, g, h) := \mathbf{E}[\mu_w(\boldsymbol{x})f(\boldsymbol{x})g(\boldsymbol{x})] - 2\mathbf{E}[\mu_z(\boldsymbol{x})h(\boldsymbol{x})] + \mathbf{E}[\ell_*^*(g(\boldsymbol{x}), h(\boldsymbol{x}))]$. Since $\mu_w$ and $\mu_z$ appear separately and linearly, $K(f, g, h)$ can be approximated by sample averages using separately labeled samples.

## 7 Experiments

In this section, we test the proposed method and compare it with baselines.

### 7.1 Data Sets

We use the following data sets for experiments.

**Synthetic data:** Features $\boldsymbol{x}$ are drawn from the two-dimensional Gaussian distribution with mean zero and covariance $10\boldsymbol{I}_2$. We set $p(y \mid \boldsymbol{x}, t)$ as the following logistic models: $p(y \mid \boldsymbol{x}, t) = 1/(1 - \exp(-y\boldsymbol{a}_t^\top \boldsymbol{x}))$, where $\boldsymbol{a}_{-1} = (10, 10)^\top$ and $\boldsymbol{a}_1 = (10, -10)^\top$. We also use the logistic models for $p_k(t \mid \boldsymbol{x})$: $p_1(t \mid \boldsymbol{x}) = 1/(1 - \exp(-tx_2))$ and $p_2(t \mid \boldsymbol{x}) = 1/(1 - \exp(-t\{x_2 + b\}))$, where $b$ is varied over 25 equally spaced points in $[0, 10]$. We investigate how the performance changes when the difference between $p_1(t \mid \boldsymbol{x})$ and $p_2(t \mid \boldsymbol{x})$ varies.

**Email data:** This data set consists of data collected in an email advertisement campaign for promoting customers to visit a website of a store [8, 27]. Outcomes are whether customers visited the website or not. We use $4 \times 5000$ and 2000 randomly sub-sampled data points for training and evaluation, respectively.

**Jobs data:** This data set consists of randomized experimental data obtained from a job training program called the National Supported Work Demonstration [17], available at `http://users.nber.org/~rdehejia/data/nswdata2.html`. There are 9 features, and outcomes are income levels after the training program. The sample sizes are 297 for the treatment group and 425 for the control group. We use $4 \times 50$ randomly sub-sampled data points for training and 100 for evaluation.

**Criteo data:** This data set consists of banner advertisement log data collected by Criteo [18] available at `http://www.cs.cornell.edu/~adith/Criteo/`. The task is to select a product to be displayed in a given banner so that the click rate will be maximized. We only use records for banners with only one advertisement slot. Each display banner has 10 features, and each product has 35 features. We take the 12th feature of a product as a treatment variable merely because it is a well-balanced binary variable. The outcome is whether the displayed advertisement was clicked. We treat the data set as the population although it is biased from the actual population since non-clicked impressions were randomly sub-sampled down to $10\%$ to reduce the data set size. We made two subsets with different treatment policies by appropriately sub-sampling according to the predefined treatment policies (see Appendix L in the supplementary material). We set $p_k(t \mid \boldsymbol{x})$ as $p_1(t \mid \boldsymbol{x}) = 1/(1 + \exp(-t\mathbf{1}^\top \boldsymbol{x}))$ and $p_2(t \mid \boldsymbol{x}) = 1/(1 + \exp(t\mathbf{1}^\top \boldsymbol{x}))$, where $\mathbf{1} := (1, \ldots, 1)^\top$.

## 7.2 Experimental Settings

We conduct experiments under the following settings.

**Methods compared:** We compare the proposed method with baselines that separately estimate the four conditional expectations in Eq. (6). In the case of binary outcomes, we use the logistic-regression-based (denoted by FourLogistic) and a neural-network-based method trained with the soft-max cross-entropy loss (denoted by FourNNC). In the case of real-valued outcomes, the ridge-regression-based (denoted by FourRidge) and a neural-network-based method trained with the squared loss (denoted by FourNNR). The neural networks are fully connected ones with two hidden layers each with 10 hidden units. For the proposed method, we use the linear-in-parameter models with Gaussian basis functions centered at randomly sub-sampled training data points (see Appendix K for more details).

**Performance evaluation:** We evaluate trained uplift models by the area under the uplift curve (AUUC) estimated on test samples with joint labels as well as *uplift curves* [26]. The uplift curve of an estimated individual uplift is the trajectory of the average uplift when individuals are gradually moved from the control group to the treated group in the descending order according to the ranking given by the estimated individual uplift. These quantities can be estimated when data are randomized experiment ones. The Criteo data are not randomized experiment data unlike other data sets, but there are accurately logged propensity scores available. In this case, uplift curves and the AUUCs can be estimated using the inverse propensity scoring [3, 20]. We conduct 50 trials of each experiment with different random seeds.

## 7.3 Results

The results on the synthetic data are summarized in Figure 1. From the plots, we can see that all methods perform relatively well in terms of AUUCs when the policies are distant from each other (i.e., $b$ is larger). However, the performance of the baseline methods immediately declines as the treatment policies get closer to each other (i.e., $b$ is smaller).[3] In contrast, the proposed method maintains its performance relatively longer until $b$ reaches the point around 2. Note that the two policies would be identical when $b = 0$, which makes it impossible to identify the individual uplift from their samples by any method since the system in Eq. (5) degenerates. Figure 2 highlights their performance in terms of the squared error. For FourNNC, test points with small policy difference $|p_1(t = 1 \mid \boldsymbol{x}) - p_2(t = 1 \mid \boldsymbol{x})|$ (colored darker) tend to have very large estimation errors. On the other hand, the proposed method has relatively small errors even for such points. Figure 3 shows results on real data sets. The proposed method and the baseline method with logistic regressors both performed better than the baseline method with neural nets on the Email data set (Figure 3a).

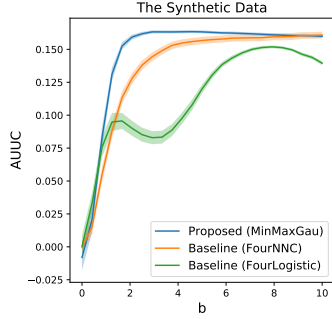

Figure 1: Results on the synthetic data. The plot shows the average AUUCs obtained by the proposed method and the baseline methods for different $b$. $p_1(t \mid \boldsymbol{x})$ and $p_2(t \mid \boldsymbol{x})$ are closer to each other when $b$ is smaller.

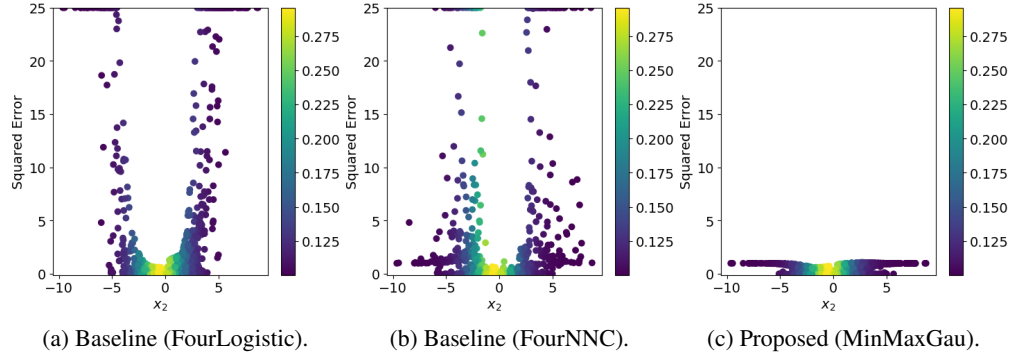

(a) Baseline (FourLogistic).    (b) Baseline (FourNNC).    (c) Proposed (MinMaxGau).

Figure 2: The plots show the squared errors of the estimated individual uplifts on the synthetic data with $b = 1$. Each point is darker-colored when $|p_1(t = 1 \mid \boldsymbol{x}) - p_2(t = 1 \mid \boldsymbol{x})|$ is smaller, and lighter-colored otherwise.

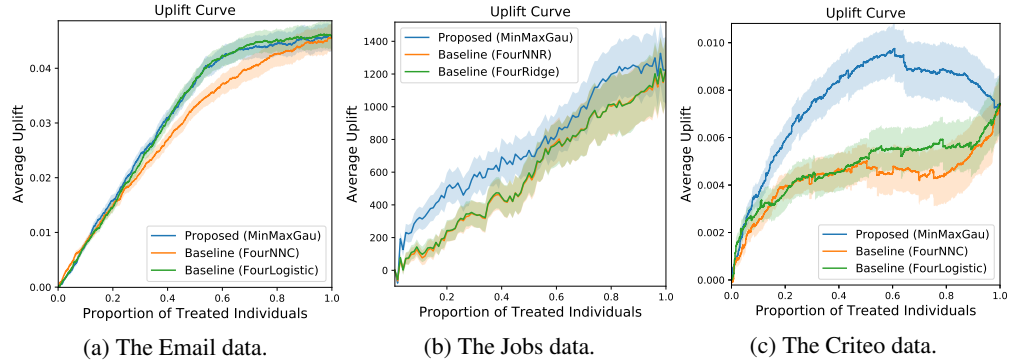

(a) The Email data.    (b) The Jobs data.    (c) The Criteo data.

Figure 3: Average uplifts as well as their standard errors on real-world data sets.

On the Jobs data set, the proposed method again performed better than the baseline methods with neural networks. For the Criteo data set, the proposed method outperformed the baseline methods (Figure 3c). Overall, we confirmed the superiority of the proposed both on synthetic and real data sets.

## 8   Conclusion

We proposed a theoretically guaranteed and practically useful method for uplift modeling or individual treatment effect estimation under the presence of systematic missing labels. The proposed method showed promising results in our experiments on synthetic and real data sets. The proposed framework is model-independent: any models can be used to approximate the individual uplift including ones tailored for specific problems and complex models such as neural networks. On the other hand, model selection may be a challenging problem due to the min-max structure. Addressing this issue would be important research directions for further expanding the applicability and improving the performance of the proposed method.

**Acknowledgments**

We are grateful to Marthinus Christoffel du Plessis and Takeshi Teshima for their inspiring suggestions and for the meaningful discussions. We would like to thank the anonymous reviewers for their helpful comments. IY was supported by JSPS KAKENHI 16J07970. JA and FY would like to thank Adway for its support. MS was supported by the International Research Center for Neurointelligence (WPI-IRCN) at The University of Tokyo Institutes for Advanced Study.

## Footnotes

[1] Among the related papers mentioned above, the most relevant one is Lewis and Syrgkanis [19], which is concurrent work with ours.

[2]$\liminf_{y \to y_0} \varphi(y) := \lim_{\delta \searrow 0} \inf_{|y - y_0| \leq \delta} \varphi(y)$.

[3]The instability of performance of FourLogistic can be explained as follows. FourLogistic uses linear models, whose expressive power is limited. The resulting estimator has small variance with potentially large bias. Since different $b$ induces different $u(\boldsymbol{x})$, the bias depends on $b$. For this reason, the method works well for some $b$ but poorly for other $b$.

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
