[Supplementary Material · supplementary.pdf]

# Appendix

## A  Average Uplift in Terms of the Individual Uplift

$$U(\pi) = \iint \sum_{t=-1,1} y p(y \mid t, \boldsymbol{x}) \pi(t \mid \boldsymbol{x}) p(\boldsymbol{x}) \mathrm{d}y \mathrm{d}\boldsymbol{x} - \iint \sum_{t=-1,1} y p(y \mid t, \boldsymbol{x}) 1[t = -1] p(\boldsymbol{x}) \mathrm{d}y \mathrm{d}\boldsymbol{x}$$

$$= \iint y [p(y \mid t = 1, \boldsymbol{x}) \pi(t = 1 \mid \boldsymbol{x}) - p(y \mid t = -1, \boldsymbol{x}) \pi(t = 1 \mid \boldsymbol{x})] p(\boldsymbol{x}) \mathrm{d}y \mathrm{d}\boldsymbol{x}$$

$$= \iint y [p(y \mid t = 1, \boldsymbol{x}) - p(y \mid t = -1, \boldsymbol{x})] \pi(t = 1 \mid \boldsymbol{x}) p(\boldsymbol{x}) \mathrm{d}y \mathrm{d}\boldsymbol{x}$$

$$= \int u(\boldsymbol{x}) \pi(t = 1 \mid \boldsymbol{x}) p(\boldsymbol{x}) \mathrm{d}\boldsymbol{x}. \tag{15}$$

## B  Area Under the Uplift Curve and Ranking

Define the following symbols:

- $C_\alpha := \Pr[f(\boldsymbol{x}) < \alpha]$,
- $U(\alpha; f) := \int u(\boldsymbol{x}) 1[\alpha \le f(\boldsymbol{x})] p(\boldsymbol{x}) \mathrm{d}\boldsymbol{x}$,
- $\mathrm{Rank}(f) := \mathbf{E}[1[f(\boldsymbol{x}') \le f(\boldsymbol{x})][u(\boldsymbol{x}') - u(\boldsymbol{x})]]$,
- $\mathrm{AUUC}(f) := \int_0^1 U(\alpha; f) \mathrm{d}C_\alpha$.

Then, we have

$$\mathrm{AUUC}(f) = \int_{-\infty}^{\infty} U(\alpha) \frac{\mathrm{d}C_\alpha}{\mathrm{d}\alpha} \mathrm{d}\alpha$$

$$= \int_{-\infty}^{\infty} U(\alpha) p_{f(\boldsymbol{x})}(\alpha) \mathrm{d}\alpha$$

$$= \int_{\mathbb{R}^d} U(f(\boldsymbol{x})) p(\boldsymbol{x}) \mathrm{d}\boldsymbol{x}$$

$$= \iint 1[f(\boldsymbol{x}) \le f(\boldsymbol{x}')] u(\boldsymbol{x}') p(\boldsymbol{x}') \mathrm{d}\boldsymbol{x}' p(\boldsymbol{x}) \mathrm{d}\boldsymbol{x}$$

$$= \mathbf{E}[1[f(\boldsymbol{x}) \le f(\boldsymbol{x}')] u(\boldsymbol{x}')]$$

$$(= \mathbf{E}[1[f(\boldsymbol{x}) \le f(\boldsymbol{x}')][y^+ - y^-]]),$$

where $y^+ \sim p(y \mid \boldsymbol{x}', t = 1)$ and $y^- \sim p(y \mid \boldsymbol{x}', t = -1)$.

Assuming $\Pr[f(\boldsymbol{x}') = f(\boldsymbol{x})] = 0$, we have

$$\mathrm{Rank}(f) := \mathbf{E}[1[f(\boldsymbol{x}) \ge f(\boldsymbol{x}')][u(\boldsymbol{x}) - u(\boldsymbol{x}')]]$$

$$= \mathbf{E}[1[f(\boldsymbol{x}) \ge f(\boldsymbol{x}')] u(\boldsymbol{x})]$$

$$- \mathbf{E}[1[f(\boldsymbol{x}) \ge f(\boldsymbol{x}')] u(\boldsymbol{x}')]$$

$$= \mathrm{AUUC}(f) - \mathbf{E}[(1 - 1[f(\boldsymbol{x}) \le f(\boldsymbol{x}')]) u(\boldsymbol{x}')]$$

$$= \mathbf{E}[u(\boldsymbol{x})] - 2 \mathrm{AUUC}(f).$$

Thus, $\mathrm{Rank}(f) = 2(\mathrm{AUUC}(f) - \mathrm{AUUC}(r))$, where $r : \mathbb{R}^d \to \mathbb{R}$ is the random ranking scoring function that outputs 1 or $-1$ with probability $1/2$ for any input $\boldsymbol{x}$. $\mathrm{Rank}(f)$ is maximized when $f(\boldsymbol{x}) \le f(\boldsymbol{x}') \iff u(\boldsymbol{x}) \le u(\boldsymbol{x}')$ for almost every pair of $\boldsymbol{x} \in \mathbb{R}^d$ and $\boldsymbol{x} \in \mathbb{R}^d$. In particular, $f = u$ is a maximizer of the objective.

## C  Proof of Lemma 1

**Lemma 1.** *For every $\boldsymbol{x}$ such that $p_1(\boldsymbol{x}) \ne p_2(\boldsymbol{x})$, $u(\boldsymbol{x})$ can be expressed as*

$$u(\boldsymbol{x}) = 2 \times \frac{\mathbf{E}_{y \sim p_1(y|\boldsymbol{x})}[y] - \mathbf{E}_{y \sim p_2(y|\boldsymbol{x})}[y]}{\mathbf{E}_{t \sim p_1(t|\boldsymbol{x})}[t] - \mathbf{E}_{t \sim p_2(t|\boldsymbol{x})}[t]}. \tag{16}$$

*Proof.*

$$\mathbf{E}_{y \sim p_1(y|\boldsymbol{x})}[y] - \mathbf{E}_{y \sim p_2(y|\boldsymbol{x})}[y] = \int \sum_{\tau=-1,1} yp(y \mid \boldsymbol{x}, t = \tau)p_1(t = \tau \mid \boldsymbol{x})\mathrm{d}y$$

$$- \int \sum_{\tau=-1,1} yp(y \mid \boldsymbol{x}, t = \tau)p_2(t = \tau \mid \boldsymbol{x})\mathrm{d}y$$

$$= \int \sum_{\tau=-1,1} yp(y \mid \boldsymbol{x}, t = \tau)(p_1(t = \tau \mid \boldsymbol{x}) - p_2(t = \tau \mid \boldsymbol{x}))\mathrm{d}y$$

$$= \sum_{\tau=-1,1} \mathbf{E}_{y \sim p(y|\boldsymbol{x},t=\tau)}[y](p_1(t = \tau \mid \boldsymbol{x}) - p_2(t = \tau \mid \boldsymbol{x}))$$

$$= \mathbf{E}_{y \sim p(y|\boldsymbol{x},t=1)}[y](p_1(t = 1 \mid \boldsymbol{x}) - p_2(t = 1 \mid \boldsymbol{x}))$$
$$+ \mathbf{E}_{y \sim p(y|\boldsymbol{x},t=-1)}[y](1 - p_1(t = 1 \mid \boldsymbol{x}) - 1 + p_2(t = 1 \mid \boldsymbol{x}))$$
$$= u(\boldsymbol{x})(p_1(t = 1 \mid \boldsymbol{x}) - p_2(t = 1 \mid \boldsymbol{x})).$$

When $p_1(t = 1 \mid \boldsymbol{x}) \neq p_2(t = 1 \mid \boldsymbol{x})$, it holds that

$$u(\boldsymbol{x}) = \frac{\mathbf{E}_{y \sim p_1(y|\boldsymbol{x})}[y] - \mathbf{E}_{y \sim p_2(y|\boldsymbol{x})}[y]}{p_1(t = 1 \mid \boldsymbol{x}) - p_2(t = 1 \mid \boldsymbol{x})}$$

$$= 2 \times \frac{\mathbf{E}_{y \sim p_1(y|\boldsymbol{x})}[y] - \mathbf{E}_{y \sim p_2(y|\boldsymbol{x})}[y]}{\mathbf{E}_{t \sim p_1(t|\boldsymbol{x})}[t] - \mathbf{E}_{t \sim p_2(t|\boldsymbol{x})}[t]}.$$

$\square$

## D   Proof of Lemma 2

**Lemma 2.** *For every $\boldsymbol{x}$ such that $p_1(\boldsymbol{x}) \neq p_2(\boldsymbol{x})$, $u(\boldsymbol{x})$ can be expressed as*

$$u(\boldsymbol{x}) = 2 \times \frac{\mathbf{E}[z \mid \boldsymbol{x}]}{\mathbf{E}[w \mid \boldsymbol{x}]},$$

*where $\mathbf{E}[z \mid \boldsymbol{x}]$ and $\mathbf{E}[w \mid \boldsymbol{x}]$ are the conditional expectations of $z$ given $\boldsymbol{x}$ over $p(z \mid \boldsymbol{x})$ and $w$ given $\boldsymbol{x}$ over $p(w \mid \boldsymbol{x})$, respectively.*

*Proof.* We have

$$\mathbf{E}[z \mid \boldsymbol{x}] = \int \zeta \left[\frac{1}{2}p_1(y = \zeta \mid \boldsymbol{x}) + \frac{1}{2}p_2(y = -\zeta \mid \boldsymbol{x})\right] \mathrm{d}\zeta$$

$$= \frac{1}{2} \int \zeta p_1(y = \zeta \mid \boldsymbol{x})\mathrm{d}\zeta + \frac{1}{2} \int \zeta p_2(y = -\zeta \mid \boldsymbol{x})\mathrm{d}\zeta$$

$$= \frac{1}{2} \int y p_1(y \mid \boldsymbol{x})\mathrm{d}y - \frac{1}{2} \int y p_2(y \mid \boldsymbol{x})\mathrm{d}y$$

$$= \frac{1}{2}\mathbf{E}_{y \sim p_1(y|\boldsymbol{x})}[y] - \frac{1}{2}\mathbf{E}_{y \sim p_2(y|\boldsymbol{x})}[y].$$

Similarly, we obtain

$$\mathbf{E}[w \mid \boldsymbol{x}] = \frac{1}{2}\mathbf{E}_{t \sim p_1(t|\boldsymbol{x})}[t] - \frac{1}{2}\mathbf{E}_{t \sim p_2(t|\boldsymbol{x})}[t].$$

Thus,

$$2 \times \frac{\mathbf{E}[z \mid \boldsymbol{x}]}{\mathbf{E}[w \mid \boldsymbol{x}]} = 2 \times \frac{\mathbf{E}_{y \sim p_1(y|\boldsymbol{x})}[y] - \mathbf{E}_{y \sim p_2(y|\boldsymbol{x})}[y]}{\mathbf{E}_{t \sim p_1(t|\boldsymbol{x})}[t] - \mathbf{E}_{t \sim p_2(t|\boldsymbol{x})}[t]} = u(\boldsymbol{x}).$$

$\square$

## E   Proof of Theorem 2

We restate Theorem 2 below.

**Theorem 2.** *Assume that $n_1 = n_2$, $\widetilde{n}_1 = \widetilde{n}_2$, $p_1(\boldsymbol{x}) = p_2(\boldsymbol{x})$, $W := \inf_{\boldsymbol{x} \in \mathcal{X}} |\mu_w(\boldsymbol{x})| > 0$, $M_{\mathcal{Z}} := \sup_{z \in \mathcal{Z}} |z| < \infty$, $M_F := \sup_{f \in F, \boldsymbol{x} \in \mathcal{X}} |f(\boldsymbol{x})| < \infty$, and $M_G := \sup_{g \in G, \boldsymbol{x} \in \mathcal{X}} |g(\boldsymbol{x})| < \infty$. Then, the following holds with probability at least $1 - \delta$ that for every $f \in F$,*

$$\mathbf{E}_{\boldsymbol{x} \sim p(\boldsymbol{x})}[(f(\boldsymbol{x}) - u(\boldsymbol{x}))^2] \leq \frac{1}{W^2} \left[\sup_{g \in G} \widehat{J}(f, g) + \mathcal{R}_{F,G}^{n,\widetilde{n}} + \left(\frac{M_z}{\sqrt{2n}} + \frac{M_w}{\sqrt{2\widetilde{n}}}\right)\sqrt{\log\frac{2}{\delta}} + \varepsilon_G(f)\right],$$

where $M_z := 4M_{\mathcal{Y}}M_{\mathcal{G}} + M_{\mathcal{G}}^2/2$, $M_w = 2M_{\mathcal{F}}M_{\mathcal{G}} + M_{\mathcal{G}}^2/2$, $\mathcal{R}_{F,G}^{n,\widetilde{n}} := 2(M_F + 4M_Z)\mathfrak{R}_{p(\boldsymbol{x},z)}^n(G) + 2(2M_F + M_G)\mathfrak{R}_{p(\boldsymbol{x},w)}^{\widetilde{n}}(F) + 2(M_F + M_G)\mathfrak{R}_{p(\boldsymbol{x},w)}^{\widetilde{n}}(G)$.

Define $J(f,g)$ and $\widehat{J}(f,g)$ as in Section 3.2 and denote

$$\varepsilon_G(f) := \sup_{g \in L^2(p)} J(f,g) - \sup_{g \in G} J(f,g).$$

**Definition 1** (Rademacher Complexity)**.** *We define the* Rademacher complexity *of $H$ over $N$ random variables following probability distribution $q$ by*

$$\mathfrak{R}_p^N(H) = \mathbf{E}_{V_1,\ldots,V_N,\sigma_1,\ldots,\sigma_N}\left[\sup_{h \in H} \frac{1}{N}\sum_{i=1}^N \sigma_i h(V_i)\right],$$

*where $\sigma_1,\ldots,\sigma_N$ are independent, $\{-1,1\}$-valued uniform random variables.*

**Lemma 3.** *Under the assumptions of Theorem 2, with probability at least $1 - \delta$, it holds that for every $f \in F$,*

$$J(f,g) \leq \widehat{J}(f,g) + \mathfrak{R}_{F,G} + \left(\frac{M_z}{\sqrt{n}} + \frac{M_w}{\sqrt{\widetilde{n}}}\right)\sqrt{\log\frac{2}{\delta}}.$$

To prove Lemma 3, we use the following lemma, which is a slightly modified version of Theorem 3.1 in Mohri et al. [22].

**Lemma 4.** *Let $H$ be a set of real-valued functions on a measurable space $\mathcal{D}$. Assume that $M := \sup_{h \in H, v \in \mathcal{D}} h(v) < \infty$. Then, for any $h \in H$ and any $\mathcal{D}$-valued i.i.d. random variables $V, V_1, \ldots, V_N$ following density $q$, we have*

$$\mathbf{E}[h(V)] \leq \frac{1}{N}\sum_{i=1}^N h(V_i) + 2\mathfrak{R}_q^N(H) + \sqrt{\frac{M^2}{N}\log\frac{1}{\delta}}. \tag{17}$$

*Proof of Lemma 4.* We follow the proof of Theorem 3.1 in Mohri et al. [22] except that we set the constant $B_\phi$ in Eq. (28) to $M/m$ when we apply McDiarmid's inequality (Section M). $\qquad\square$

Now, we prove Lemma 3.

*Proof of Lemma 3.* For any $f \in \mathcal{F}$, $g \in \mathcal{G}$, $\boldsymbol{x}', \widetilde{\boldsymbol{x}}' \in \mathcal{X}$, $z' \in \mathcal{Z} := \{y, -y \mid y \in \mathcal{Y}\}$, and $w' \in \{-1,1\}$, we define $h_z$ and $h_w$ as follows:

$$h_z(\boldsymbol{x}', z'; g) := -4z'g(\boldsymbol{x}') - \frac{1}{2}g(\boldsymbol{x}')^2,$$

$$h_w(\widetilde{\boldsymbol{x}}', w'; f, g) := w'f(\widetilde{\boldsymbol{x}}')g(\widetilde{\boldsymbol{x}}') - \frac{1}{2}g(\widetilde{\boldsymbol{x}}')^2.$$

Denoting $H_z := \{(\boldsymbol{x}', z') \mapsto h_z(\boldsymbol{x}', z'; g) \mid g \in G\}$, we have

$$\sup_{h \in H_z, \boldsymbol{x}' \in \mathcal{X}, z' \in \mathcal{Z}}\left|h(\boldsymbol{x}', z')\right| \leq 4M_Z M_G + \frac{1}{2}M_G^2 =: M_z < \infty,$$

and thus, we can apply Lemma 4 to confirm that with probability at least $1 - \delta/2$,

$$\mathbf{E}_{(\boldsymbol{x},z) \sim p(\boldsymbol{x},z)}[h_z(\boldsymbol{x}, z; g)] \leq \frac{1}{n}\sum_{(\boldsymbol{x}_i,z_i) \in S_z} h_z(\boldsymbol{x}_i, z_i; g) + 2\mathfrak{R}_p^n(H_z) + \sqrt{\frac{M_z^2}{n}\log\frac{2}{\delta}},$$

where $\{(\boldsymbol{x}_i, z_i)\}_{i=1}^n =: S_z$ are the samples defined in Section 4.1. Similarly, it holds that with probability at least $1 - \delta/2$,

$$\mathbf{E}_{(\widetilde{\boldsymbol{x}},w) \sim p(\boldsymbol{x},w)}[h_w(\widetilde{\boldsymbol{x}}, w; f, g)] \leq \frac{1}{\widetilde{n}}\sum_{(\widetilde{\boldsymbol{x}},w_i) \in S_w} h_w(\widetilde{\boldsymbol{x}}_i, w_i; f, g) + 2\mathfrak{R}_p^{\widetilde{n}}(H_w) + \sqrt{\frac{M_w^2}{\widetilde{n}}\log\frac{2}{\delta}},$$

where $H_w := \{(\widetilde{\boldsymbol{x}}', w') \mapsto h_w(\widetilde{\boldsymbol{x}}', w'; f, g) \mid f \in F, g \in G\}$, $M_w := M_F M_G + M_G^2/2$, and $\{(\widetilde{\boldsymbol{x}}_i, w_i)\}_{i=1}^n =: S_w$ are the samples defined in Section 4.1. By the union bound, we have the following with probability at least $1 - \delta$:

$$\mathbf{E}_{(\boldsymbol{x},z) \sim p(\boldsymbol{x},z)}[h_z(\boldsymbol{x}, z; g)] + \mathbf{E}_{(\widetilde{\boldsymbol{x}},w) \sim p(\boldsymbol{x},w)}[h_w(\widetilde{\boldsymbol{x}}, w; f, g)] \tag{18}$$

$$\leq \frac{1}{n}\sum_{(\boldsymbol{x}_i,z_i) \in S_z} h_z(\boldsymbol{x}_i, z_i, g) + \frac{1}{\widetilde{n}}\sum_{(\widetilde{\boldsymbol{x}},w_i)} h_w(\boldsymbol{x}_i, w_i, f, g) \tag{19}$$

$$+ 2(\mathfrak{R}_p^n(H_z) + \mathfrak{R}_p^{\widetilde{n}}(H_w)) + \left(\frac{M_z}{\sqrt{n}} + \frac{M_w}{\sqrt{\widetilde{n}}}\right)\sqrt{\log\frac{2}{\delta}}, \tag{20}$$

Using some properties of the Rademacher complexity including Talagrand's lemma, we can show that

$$\mathfrak{R}_p^n(H_z) \leq (M_F + 4M_Z)\mathfrak{R}_p^n(G), \tag{21}$$

$$\mathfrak{R}_p^{\widetilde{n}}(H_w) \leq (2M_F + M_G)\mathfrak{R}_p^{\widetilde{n}}(F) + (M_F + M_G)\mathfrak{R}_p^{\widetilde{n}}(G). \tag{22}$$

Clearly,

$$\widehat{J}(f,g) = \frac{1}{n}\sum_{(\boldsymbol{x}_i, z_i) \in S_z} h(\boldsymbol{x}_i, z_i; g) + \frac{1}{\widetilde{n}}\sum_{(\widetilde{\boldsymbol{x}}_i, w_i) \in S_w} h(\widetilde{\boldsymbol{x}}_i, w_i; f, g),$$

$$J(f,g) = \mathbf{E}_{(\boldsymbol{x}, z) \sim p(\boldsymbol{x}, z)}[h_z(\boldsymbol{x}, z; g)] + \mathbf{E}_{(\widetilde{\boldsymbol{x}}, w) \sim p(\boldsymbol{x}, z)}[h_w(\widetilde{\boldsymbol{x}}, w; f, g)].$$

From Eq. (20), Eq. (21), and Eq. (22), we obtain

$$J(f,g) \leq \widehat{J}(f,g) + \mathfrak{R}_{F,G} + \left(\frac{M_z}{\sqrt{n}} + \frac{M_w}{\sqrt{\widetilde{n}}}\right)\sqrt{\log\frac{2}{\delta}}, \tag{23}$$

where

$$\mathfrak{R}_{F,G} := 2(M_F + 4M_Z)\mathfrak{R}_p^n(G) + 2(2M_F + M_G)\mathfrak{R}_p^{\widetilde{n}}(F) + 2(M_F + M_G)\mathfrak{R}_p^{\widetilde{n}}(G).$$

$\square$

Finally, we prove Theorem 2.

*Proof of Theorem 2.* From Lemma 3, with probability at least $1 - \delta$, it holds that for all $f \in F$

$$\sup_{g \in G} J(f,g) \leq \sup_{g \in G} \widehat{J}(f,g) + \mathfrak{R}_{F,G} + \left(\frac{M_z}{\sqrt{n}} + \frac{M_w}{\sqrt{\widetilde{n}}}\right)\sqrt{\log\frac{2}{\delta}}. \tag{24}$$

Moreover, recalling $W := \inf_{\boldsymbol{x} \in \mathcal{X}} |\mu_w(\boldsymbol{x})|$ and $\sup_{g \in L^2(p)} J(f,g) = \mathbf{E}[(\mu_w(\boldsymbol{x})f(\boldsymbol{x}) - \mu_z(\boldsymbol{x}))^2]$, we have

$$\mathbf{E}\left[(f(\boldsymbol{x}) - u(\boldsymbol{x}))^2\right] = \mathbf{E}\left[\left(f(\boldsymbol{x}) - \frac{\mu_z(\boldsymbol{x})}{\mu_w(\boldsymbol{x})}\right)^2\right] \tag{25}$$

$$\leq \frac{1}{W^2}\mathbf{E}[(\mu_w(\boldsymbol{x})f(\boldsymbol{x}) - \mu_z(\boldsymbol{x}))^2] \tag{26}$$

$$= \frac{1}{W^2}\left[\varepsilon_G(f) + \sup_{g \in G} J(f,g)\right]. \tag{27}$$

Combining Eq. (24) and Eq. (27) yields the inequality of the theorem. $\square$

## F Proof of Corollary 1

**Corollary 1.** *Let* $F = \{x \mapsto \boldsymbol{\alpha}^\top \boldsymbol{\phi}(\boldsymbol{x}) \mid \|\boldsymbol{\alpha}\|_2 \leq \Lambda_F\}$, $G = \{x \mapsto \boldsymbol{\beta}^\top \boldsymbol{\psi}(\boldsymbol{x}) \mid \|\boldsymbol{\beta}\|_2 \leq \Lambda_G\}$, *and assume that* $r_F := \sup_{\boldsymbol{x} \in \mathcal{X}} \|\boldsymbol{\phi}(\boldsymbol{x})\|_2 < \infty$ *and* $r_G := \sup_{\boldsymbol{x} \in \mathcal{X}} \|\boldsymbol{\psi}(\boldsymbol{x})\|_2 < \infty$, *where* $\|\cdot\|_2$ *is the* $L_2$-*norm. Under the assumptions of Theorem 2, it holds with probability at least* $1 - \delta$ *that for every* $f \in F$,

$$\mathbf{E}_{\boldsymbol{x} \sim p(\boldsymbol{x})}[(f(\boldsymbol{x}) - u(\boldsymbol{x}))^2] \leq \frac{1}{W^2}\left[\sup_{g \in G} \widehat{J}(f,g) + \frac{C_z\sqrt{\log\frac{2}{\delta}} + D_z}{\sqrt{2n}} + \frac{C_w\sqrt{\log\frac{2}{\delta}} + D_w}{\sqrt{2\widetilde{n}}} + \varepsilon_G(f)\right],$$

*where* $C_z := r_G^2\Lambda_G^2 + 4r_G\Lambda_G M_{\mathcal{Y}}$, $C_w := 2r_F^2\Lambda_F^2 + 2r_F r_G \Lambda_F \Lambda_G + r_G^2\Lambda_G^2$, $D_z := r_G^2\Lambda_G^2/2 + 4r_G\Lambda_G M_{\mathcal{Y}}$, *and* $D_w := r_G^2\Lambda_G^2/2 + 4r_F r_G \Lambda_F \Lambda_G$.

*Proof.* Under the assumptions, it is known that the Rademacher complexity of the linear-in-parameter model $F$ can be upper bounded as follows [22]:

$$\mathfrak{R}_p^N(F) \leq \frac{r_F\Lambda_F}{\sqrt{N}}.$$

We can bound $\mathfrak{R}_p^N(G)$ similarly. Applying these bounds to Theorem 2, we obtain the statement of Corollary 1. $\square$

## G    Proof of Theorem 3

We prove the following, formal version of Theorem 3.

**Theorem 3.** *Under the assumptions of Corollary 1, it holds with probability at least $1 - \delta$ that $\mathbf{E}[(\widehat{f}(\boldsymbol{x}) - u(\boldsymbol{x}))^2] \leq (4e_{n,\delta} + 2\varepsilon_G^F + \varepsilon_F)/W^2$, where $\varepsilon_G^F := \sup_{f \in F} \varepsilon_G(f)$, and $\varepsilon_F := \inf_{f \in F} J(f)$, $\widehat{f} \in F$ is any approximate solution to $\inf_{f \in F} \sup_{g \in G} \widehat{J}(f, g)$ satisfying $\sup_{g \in G} \widehat{J}(\widehat{f}, g) \leq \inf_{f \in F} \sup_{g \in G} \widehat{J}(f, g) + e_{n,\delta}$, and*

$$e_{n,\delta} := \frac{C_z \sqrt{\log \frac{2}{\delta}} + D_z}{\sqrt{2n}} + \frac{C_w \sqrt{\log \frac{2}{\delta}} + D_w}{\sqrt{2\widetilde{n}}}.$$

*Proof.* Let $J(f) := \sup_{g \in L^2} J(f, g) = \mathbf{E}[(\mu_w(\boldsymbol{x})f(\boldsymbol{x}) - \mu_z(\boldsymbol{x}))^2]$, $J_G(f) := \sup_{g \in G} J(f, g)$, $\widehat{J}_G(f) := \sup_{g \in G} \widehat{J}(f, g)$. Let $\widetilde{f} \in F$ be any approximate solution to $\inf_{f \in F} J(f)$ satisfying $J(\widetilde{f}) \leq \varepsilon_F + e_{n,\delta}$.

As a special case of Eq. 24, we can prove that with probability at least $1 - \delta$, it holds for every $f \in F$ that $J_G(f) \leq \widehat{J}_G(f) + e_{n,\delta}$. From Corollary 1, it holds that with probability at least $1 - \delta$,

$$
\begin{aligned}
J(\widehat{f}) &\leq \left[ J(\widehat{f}) - J_G(\widehat{f}) \right] + \left[ J_G(\widehat{f}) - \widehat{J}_G(\widehat{f}) \right] + \left[ \widehat{J}_G(\widehat{f}) - \widehat{J}_G(\widetilde{f}) \right] \\
&\quad + \left[ \widehat{J}_G(\widetilde{f}) - J_G(\widetilde{f}) \right] + \left[ J_G(\widetilde{f}) - J(\widetilde{f}) \right] + J(\widetilde{f}) \\
&\leq \varepsilon_G^F + e_{n,\delta} + e_{n,\delta} \\
&\quad + e_{n,\delta} + \varepsilon_G^F + [\varepsilon_F + e_{n,\delta}] \\
&\leq 4e_{n,\delta} + 2\varepsilon_G^F + \varepsilon_F.
\end{aligned}
$$

Since $\mathbf{E}[(\widehat{f}(\boldsymbol{x}) - u(\boldsymbol{x}))^2] \leq \frac{1}{W^2} J(\widehat{f})$, we obtain the bound in Theorem 3.   □

## H    Binary Outcomes

When outcomes $y$ take on binary values, e.g., success or failure, without loss of generality, we can assume that $y \in \{-1, 1\}$. Then, by the definition of the individual uplift, $u(\boldsymbol{x}) \in [-2, 2]$ for any $\boldsymbol{x} \in \mathbb{R}^d$. In order to incorporate this fact, we may add the following range constraints on $f$: $-2 \leq f(\boldsymbol{x}) \leq 2$ for every $\boldsymbol{x} \in \{\boldsymbol{x}_i\}_{i=1}^n \cup \{\widetilde{\boldsymbol{x}}_i\}_{i=1}^{\widetilde{n}}$.

## I    Cases Where $p_1(\boldsymbol{x}) \neq p_2(\boldsymbol{x})$ or $(n_1, \widetilde{n}_1) \neq (n_1, \widetilde{n}_1)$

So far, we have assumed that $p_1(\boldsymbol{x}) = p_2(\boldsymbol{x})$, $m_1 = m_2$, and $n_1 = n_2$. The proposed method can be adapted to the more general case where these assumptions may not hold.

Let $r_k(\boldsymbol{x}) = \frac{n}{2n_k} \cdot \frac{p(\boldsymbol{x})}{p_k(\boldsymbol{x})}$ and $\widetilde{r}_k(\boldsymbol{x}) = \frac{\widetilde{n}}{2\widetilde{n}_k} \cdot \frac{p(\boldsymbol{x})}{p_k(\boldsymbol{x})}$, $k = 1, 2$, for every $\boldsymbol{x}$ with $p_k(\boldsymbol{x}) > 0$. Let $k_i := 1$ if the sample $\boldsymbol{x}_i$ originally comes from $p_1(\boldsymbol{x})$, and $k_i := 2$ if it comes from $p_2(\boldsymbol{x})$. Similarly, define $\widetilde{k}_i \in \{1, 2\}$ according to whether $\widetilde{\boldsymbol{x}}_i$ comes from $p_1(\boldsymbol{x})$ or $p_2(\boldsymbol{x})$. Then, unbiased estimators of the three terms in the proposed objective Eq. (10) are given as the following weighted sample averages using $r_k$ and $\widetilde{r}_k$:

$$\mathbf{E}_{\boldsymbol{x} \sim p(\boldsymbol{x})}[wf(\boldsymbol{x})g(\boldsymbol{x})] \approx \frac{1}{\widetilde{n}} \sum_{i=1}^{\widetilde{n}} [w_i f(\widetilde{\boldsymbol{x}}_i) g(\widetilde{\boldsymbol{x}}_i) \widetilde{r}_{\widetilde{k}_i}(\widetilde{\boldsymbol{x}}_i)],$$

$$\mathbf{E}_{\boldsymbol{x} \sim p(\boldsymbol{x})}[zg(\boldsymbol{x})] \approx \frac{1}{n} \sum_{i=1}^{n} [z_i g(\boldsymbol{x}_i) r_{k_i}(\boldsymbol{x}_i)]$$

$$\mathbf{E}_{\boldsymbol{x} \sim p(\boldsymbol{x})}[g(\boldsymbol{x})^2] \approx \frac{1}{2n} \sum_{i=1}^{n} [g(\boldsymbol{x}_i)^2 r_{k_i}(\boldsymbol{x}_i)] + \frac{1}{2\widetilde{n}} \sum_{i=1}^{\widetilde{n}} [g(\widetilde{\boldsymbol{x}}_i)^2 \widetilde{r}_{\widetilde{k}_i}(\widetilde{\boldsymbol{x}}_i)].$$

The density ratios $p_k(\boldsymbol{x})/p(\boldsymbol{x})$ can be accurately estimated using i.i.d. samples from $p_k(\boldsymbol{x})$ and $p(\boldsymbol{x})$ [21, 23, 35, 38].

## J    Unbiasedness of the Weighted Sample Average

Below, we show that the weighted sample averages are unbiased estimates. We only prove for $\mathbf{E}[wf(\boldsymbol{x})g(\boldsymbol{x})]$ since the other cases can be proven similarly. The expectation of the weighted sample average transforms as

follows:

$$\frac{1}{\widetilde{n}} \sum_{i=1}^{\widetilde{n}} \mathbf{E}_{\widetilde{\boldsymbol{x}}_i^{(k)} \sim p_k(\boldsymbol{x}), t_i^{(k)} \sim p_k(t|\widetilde{\boldsymbol{x}}_i^{(k)})} \left[ w_i f(\widetilde{\boldsymbol{x}}_i) g(\widetilde{\boldsymbol{x}}_i) \widetilde{r}_{\widetilde{k}_i}(\widetilde{\boldsymbol{x}}_i) \right]$$

$$= \frac{1}{\widetilde{n}} \sum_{k=1,2} \sum_{i=1}^{\widetilde{n}_k} \mathbf{E}_{\boldsymbol{x} \sim p_k(\boldsymbol{x}), t \sim p_k(t|\boldsymbol{x})} \left[ (-1)^{k-1} t f(\boldsymbol{x}) g(\boldsymbol{x}) \frac{\widetilde{n}}{2\widetilde{n}_k} \cdot \frac{p(\boldsymbol{x})}{p_k(\boldsymbol{x})} \right]$$

$$= \frac{1}{2} \sum_{k=1,2} \mathbf{E}_{\boldsymbol{x} \sim p(\boldsymbol{x}), t \sim p_k(t|\boldsymbol{x})} \left[ (-1)^{k-1} t f(\boldsymbol{x}) g(\boldsymbol{x}) \right]$$

$$= \iint (-1)^{k-1} t \sum_{k=1,2} \frac{1}{2} p_k(t \mid \boldsymbol{x}) f(\boldsymbol{x}) g(\boldsymbol{x}) p(\boldsymbol{x}) \mathrm{d}t \mathrm{d}\boldsymbol{x}$$

$$= \iint w p(w \mid \boldsymbol{x}) f(\boldsymbol{x}) g(\boldsymbol{x}) p(\boldsymbol{x}) \mathrm{d}t \mathrm{d}\boldsymbol{x}$$

$$= \mathbf{E}_{\boldsymbol{x} \sim p(\boldsymbol{x}), w \sim p(w|\boldsymbol{x})} [w f(\boldsymbol{x}) g(\boldsymbol{x})].$$

## K  Gaussian Basis Functions Used in Experiments

The $l$-th element of $\boldsymbol{\phi}(\boldsymbol{x}) = (\phi_1(\boldsymbol{x}), \dots, \phi_{b_f}(\boldsymbol{x}))^\top$ is defined by

$$\phi_l(\boldsymbol{x}) := \exp\left( \frac{-\left\| \boldsymbol{x} - \boldsymbol{x}^{(l)} \right\|^2}{\sigma^2} \right),$$

where $\boldsymbol{x}^{(l)}$, $l = 1, \dots, b_f$, are randomly chosen training data points. We used $b_f = 100$ and $\sigma = 25$ for all experiments. $\boldsymbol{\psi}$ is defined similarly.

## L  Justification of the Sub-Sampling Procedure

Suppose that we want a sample subset $S_k$ following the treatment policy $p_k(t \mid \boldsymbol{x})$. For each sample $(\boldsymbol{x}_i, t_i, y_i) \sim p(\boldsymbol{x}, t, y)$ in the original dataset, we randomly add it into $S_k$ with probability proportional to $p_k(t_i \mid \boldsymbol{x}_i)/p(t_i \mid \boldsymbol{x}_i)$. Then,

$$p(\boldsymbol{x}_i, t_i, y_i \mid (\boldsymbol{x}_i, t_i, y_i) \in S_k) = \frac{p((\boldsymbol{x}_i, t_i, y_i) \in S_k \mid \boldsymbol{x}_i, t_i, y_i) p(\boldsymbol{x}_i, t_i, y_i)}{\int \sum_{y_i, t_i} p((\boldsymbol{x}_i, t_i, y_i) \in S_k \mid \boldsymbol{x}_i, t_i, y_i) p(\boldsymbol{x}_i, t_i, y_i) \mathrm{d}\boldsymbol{x}_i}$$

$$= \frac{p_k(t_i \mid \boldsymbol{x}_i) p(y_i \mid \boldsymbol{x}_i, t_i) p(\boldsymbol{x}_i)}{\int \sum_{y_i, t_i} p_k(t_i \mid \boldsymbol{x}_i) p(y_i \mid \boldsymbol{x}_i, t_i) p(\boldsymbol{x}_i) \mathrm{d}\boldsymbol{x}_i}$$

$$= p_k(t_i \mid \boldsymbol{x}_i) p(y_i \mid \boldsymbol{x}_i, t_i) p(\boldsymbol{x}_i).$$

This means that the subsamples $S_k$ preserve the original $p(y \mid \boldsymbol{x}, t)$ and $p(\boldsymbol{x})$ but follow the desired treatment policy $p_k(t \mid \boldsymbol{x})$.

## M  McDiarmid's Inequality

Although McDiarmid's inequality is a well known theorem, we present the statement to make the paper self-contained.

**Theorem 4** (McDiarmid's inequality). *Let $\varphi : \mathcal{D}^N \to \mathbb{R}$ be a measurable function. Assume that there exists a real number $B_\varphi > 0$ such that*

$$\left| \varphi(v_1, \dots, v_N) - \varphi(v_1', \dots, v_N') \right| \leq B_\varphi, \tag{28}$$

*for any $v_i, \dots, v_N, v_1, \dots, v_N' \in \mathcal{D}$ where $v_i = v_i'$ for all but one $i \in \{1, \dots, N\}$. Then, for any $\mathcal{D}$-valued independent random variables $V_1, \dots, V_N$ and any $\delta > 0$ the following holds with probability at least $1 - \delta$:*

$$\varphi(V_1, \dots, V_N) \leq \mathbf{E}[\varphi(V_1, \dots, V_N)] + \sqrt{\frac{B_\varphi^2 N}{2} \log \frac{1}{\delta}}.$$