[Reviews · NeurIPS 2018]

Reviewer 1



This paper proposes an approach to heterogeneous treatment effect estimation (what it calls “uplift modeling”) from separate populations. A simple version of the setup of this paper is as follows. We have two populations, k = 1, 2, with different probabilities of treatment conditional on observed features, Pk[T|X] (the paper also allows for the case where these need to be estimated). We have access to covariate-outcome pairs (X, Y) drawn from both populations, so we can estimate Ek[Y|X]. We assume potential outcomes Y(-1), Y(1), and assume that E[Y(T) | X] doesn’t depend on setup k. What we would really want is to estimate a conditional average treatment effect tau(x) = E[Y(1) - Y(-1) | X = x]. However, we don’t observe assigned treatments for each sample, i.e., we don’t see (X, T, Y); rather we only know marginal treatment probabilities Pk[T|X] along with data (X, Y). The main result of the paper is that we can still identify tau(x) in terms of observable moments, (A) tau(x) = 2 (E2[Y|X=x] - E1[Y|X=x]) / (E2[T|X=x] - E1[T|X=x]), and it then derives estimators for tau(x). Overall, I found this to be a fascinating paper, in that it states a simple, seemingly difficult problem, and pushes quite far in its results. However, it appears that the setup discussed in this paper is closely related to instrumental variables regression, and that the authors have essentially re-derived (fairly sophisticated forms of) two-stage least-squares. As background, in instrumental variables regression, we usually observe an instrument Z that “nudges” individuals towards treatment; we then observe a tuple (X, Y, T, Z). There is a concern, however, that the treatment T itself may not be unconfounded. The solution, then, is to use the effect of Z on Y to identify the causal effect of T (because we assume that Z nudges on T in a random way), resulting in an identification result: (B) tau(x) = 2 (E[Y|X=x, Z=2] - E[Y|X=x, Z=1]) / (E[T|X=x, Z=2] - E[T|X=x, Z=1]). Notice that (A) and (B) are exactly the same, provided we use “setup” in case (A) as our instrument Z in case (B). Several methods for heterogeneous treatment effect estimation via instrumental variables have recently been considered, including Athey, Susan, Julie Tibshirani, and Stefan Wager. "Generalized random forests." arXiv preprint arXiv:1610.01271 (2016). Lewis, Greg, and Vasilis Syrgkanis. "Adversarial Generalized Method of Moments." arXiv preprint arXiv:1803.07164 (2018). Hartford, Jason, Greg Lewis, Kevin Leyton-Brown, and Matt Taddy. "Deep iv: A flexible approach for counterfactual prediction." In International Conference on Machine Learning, pp. 1414-1423. 2017. For a review of fundamental results, see Imbens, Guido W. "Instrumental Variables: An Econometrician’s Perspective." Statistical Science 29.3 (2014): 323-358. The approach presented here (based on solving a min-max optimization problem) appears to be novel, and potentially interesting, in the context of the existing literature. However, because the paper seems unaware of its nearest neighbors in the literature, it is very difficult to accurately assess its merit. Thus, although I enjoyed reading this paper, I am now voting for rejection. However, this should really be taken as an encouragement for the authors to re-visit their research question in light of the IV literature, and then resubmit to ICML or JMLR. Furthermore, I think this paper has potential for considerable impact; but as it’s currently written, it seems to mostly just target a niche problem in marketing, and so would probably not get the readership it deserves even if published at NIPS. Relatedly, for broader scientific impact, I’d recommend the authors use nomenclature around heterogeneous treatment effect estimation rather than uplift (which is, as far as I can tell, a term only used in marketing). Some recent references include: Hill, Jennifer L. "Bayesian nonparametric modeling for causal inference." Journal of Computational and Graphical Statistics 20.1 (2011): 217-240. Imai, Kosuke, and Marc Ratkovic. "Estimating treatment effect heterogeneity in randomized program evaluation." The Annals of Applied Statistics 7.1 (2013): 443-470. Künzel, Sören R., et al. "Meta-learners for Estimating Heterogeneous Treatment Effects using Machine Learning." arXiv preprint arXiv:1706.03461 (2017). Wager, Stefan, and Susan Athey. "Estimation and Inference of Heterogeneous Treatment Effects using Random Forests." arXiv preprint arXiv:1510.04342 (2015).

Reviewer 2



Summary: The paper introduces a new method for estimating personalized treatment effects (i.e. individual level effects based on policies that are functions of relevant covariates) from data obtained under two differing policies where for each record either the treatment or the target effect has been recorded, but never simultaneously, e.g. due to privacy reasons. It is shown the optimal policy can be recovered from the available data (assuming some appropriate functional form is known), but that a naive approach (estimating 4 required quantities separately from the data) may result in unstable solutions. Instead an alternative method, called MinMaxGauss, is proposed that rephrases the target optimal policy as the solution to an empirical optimization problem that should be more robust. Error bounds based on Rademacher complexity are derived for the difference between the inferred and optimal policy, and more general loss functions are introduced. The resulting method is compared against two baseline (‘naive’) approaches on both synthetic and real world data sets. The problem is well known, and frequently encountered in areas where it is difficult to track the impact of different strategies due to privacy concerns.The fact that it is still possible to infer the treatment effect even though these are never measured together may initially be surprising, but the key to the solution: having data from two different policies, is closely related to recent insights into multi-source causal discovery. However, the fact that this also suffices to determine *individual* treatment policies was (at least to me) a new and unexpected find. The reformulation in terms of a regularized empirical optimization problem is both elegant and flexible, and the resulting error bounds are very useful and impressive (although I did not check the derivation of Thm2+Cor1). Regarding novelty: I checked, but as I am no expert in uplift modelling I am not entirely certain to what extent there may be overlap with existing articles/methods. Still, as far as I can tell the solution presented is original, interesting, and significant. The paper itself is well written, and makes the entire problem setting and solution good to follow even for relative novices into uplift modelling. The only disappointment / source of concern was the evaluation part: experiments+results were strangely messy (see details below) and in some cases the figures presented seemed to contradict the conclusions drawn. I suspect this is partly due to some unfortunate textual/caption mistakes, but it does definitely leave room for improvement in order to fully vindicate the claim of superior efficacy of the method over the naive baseline approaches. (That said: as far as I can tell even the baseline methods are new, so in that sense even those are already relevant). In summary: challenging problem, interesting solution, good paper => accept. ======= Details: 42 : typo ‘have exposed’ 60: so treatment is entirely mediated by variables x? 61: typo: repeat of case 1 (instead of -1) 115: ‘by multiplying y to Eq.(5) …’ => bit unclear 143: repeat ’n_1 = n_2’ 170: function’f’ is clear, but can you give an intuitive interpretation for the function ‘g’ ? 189+,Th2,Cor1: interesting result, though I did not check this derivation .. 229: ‘point in [0,2]’ => in Fig1a ‘b’ seems to run from [0,10] ? 243/4: ‘impression’? ; also ’has 10 features … we take the 12th’? 248: ‘we can virtually shift ..’ => this sounds a bit suspicious: not sure whether I agree Fig1a: is this correct? is suggests the FourNNC baseline outperforms the proposed method … and does not seem to match Fig1b … idem: why the temporary breakdown of FourLogistic for values of b [1.5,3]? Fig1b/c: what is the coloured axis? .. also: text suggest ‘light = low’, axis the opposite? Fig.2: use consistent colours for the methods between the plots idem, Fig2b: what method is ‘MinMaxLin’? (not mentioned in text) 286: ‘ … problem, …’ => remove comma (alters the meaning of the sentence to one that is definitely wrong) 290-293: this is a very relevant issue that should probably be part of section 5

Reviewer 3



UPDATE: I have read the authors' reply, and am satisfied with the way it has addressed the concerns I had. This paper introduces the problem named in the title, where uplift (individual treatment effect) is to be estimated from data in which either the treatment or the outcome is missing for each data point. By a clever rewriting of the uplift, it can be estimated in one go, giving better performance than naive methods that would need to estimate four different functions. The problem addressed is significant, and the proposed method is novel to my knowledge. The article is clear and well-written. I hope the authors can address the following concerns: - The assumption (2) currently states that for all x_0, the probability of treatment must be different between p_1 and p_2 for that x. This is significantly stronger than "that the treatment policies are distinct" as you write in the text. Which do you mean? - In the statement of Lemma 1, based on its proof and on assumption (2), I would expect the condition to be p_1(t|x) \neq p_2(t|x) here. - Figure 1(a) displays results for b from 0 to 10, while according to the text, b is in [0,2]. More importantly, the plot shows that the baseline FourNCC outperforms the proposed method for almost all b, but this is not mentioned in the text. Please clarify if I'm misunderstanding something here, or otherwise change the text everywhere to reflect that the proposed method does not always outperform the baseline(s). - Proof of Lemma 1: In the big system, I think the expectation in the third line from the bottom should read E_{...} [y] (p1... - p2...), similar to the line below it. Minor comments: line 42: have exposed -> were exposed l61: the second equation should have -1's instead of 1's l143: the second n_1 = n_2 should have ~'s l165: separately -> are separately l187: densiy -> density l205: upto -> [down?] to l247: upto -> down to l286: in "problem, known as", remove the comma: with it, this sentence means something very different! l385: \ missing